# Serum Bilirubin and Sperm Quality in Adult Population

**DOI:** 10.3390/toxics10060295

**Published:** 2022-05-30

**Authors:** Yuan-Yuei Chen, Wei-Liang Chen

**Affiliations:** 1Department of Pathology, Tri-Service General Hospital, School of Medicine, National Defense Medical Center, Taipei 114, Taiwan; fu84fu840618@gmail.com; 2Department of Pathology, Tri-Service General Hospital Songshan Branch, School of Medicine, National Defense Medical Center, Taipei 114, Taiwan; 3Division of Family Medicine, Department of Family and Community Medicine, Tri-Service General Hospital, School of Medicine, National Defense Medical Center, Taipei 114, Taiwan; 4Division of Geriatric Medicine, Department of Family and Community Medicine, Tri-Service General Hospital, School of Medicine, National Defense Medical Center, Taipei 114, Taiwan; 5Department of Biochemistry, National Defense Medical Center, Taipei,114, Taiwan

**Keywords:** serum bilirubin, sperm quality, adult population

## Abstract

The neurotoxicity of bilirubin has been extensively reported in numerous studies. However, the association between bilirubin and male fertility has not yet been studied. The main goal of this study was to investigate the association between serum total bilirubin and sperm quality in an adult population. In this cross-sectional study, 9057 participants who attended the MJ health examination (2010–2016) were enrolled. Sperm specimens were collected by masturbation, and sperm quality was analyzed in accordance with the WHO criteria. Serum total bilirubin levels were measured by an automatic biochemical profile analyzer. Thereafter, the associations between serum total bilirubin and sperm quality were determined by a multivariable linear regression. Serum total bilirubin was inversely associated with sperm concentration and normal morphology with β values of −13.82 (95% CI: −26.99, −0.64) and −18.38 (95% CI: −30.46, −6.29) after adjusting for covariables. The highest levels of serum total bilirubin were significantly associated with sperm concentration and normal morphology with β values of −14.15 (95% CI: −28.36, 0.06) and −21.15 (95% CI: −33.99, −8.30). Our study highlighted the potential impact of serum bilirubin on sperm quality in a male population. Additional longitudinal research is necessary to explore these findings and underlying mechanisms.

## 1. Introduction

Bilirubin, a tetrapyrrolic pigment and albumin-bound reversible compound, is found in plasma [1]. It is a derivative of heme catabolism, which is released by hemoglobin and cytochromes [2]. Serum bilirubin levels represent the condition of the heme turnover rate, canalicular excretion, and hepatic uptake and conjugation [3]. Numerous studies have reported that hyperbilirubinemia may potentially lead to irreversible neurological damage by accumulating bilirubin in the central nervous system [4,5,6]. Unbound bilirubin induces a variety of cellular and molecular events that result in neurotoxicity [7,8].

Mounting studies have indicated that decreasing sperm quality is associated with multiple systemic diseases. Emerging evidence has reported that obesity negatively impacts male fertility and directly changes sperm function and molecular composition [9,10,11]. Metabolic syndrome, a cluster of medical conditions characterized by abdominal obesity, dyslipidemia, hypertension, and high fasting glucose, is suggested to be associated with alteration of spermatogenesis [12,13]. Reduced levels of serum testosterone and sperm quality have been found in patients with nonalcoholic fatty liver disease (NAFLD) [14]. However, the relationship between bilirubin and male reproductive function has not yet been examined. The goal of the current study was to investigate the relationship between serum total bilirubin and sperm quality in an adult population from Taiwan.

## 2. Methods

### 2.1. Study Design and Participants

The MJ Health Center is a membership-oriented private institute with four health check-up clinics in Taiwan, the MJ Health Management Institution. The center provides periodic health examinations to its members. A series of tests such as blood, urine, anthropometric measurements, physical examination, and medical history are included in this large health research database.

In this study, we excluded those who had missing data on serum total bilirubin, sperm quality analysis, and demographic characteristic data. A total of 9057 eligible participants who attended one or more health examinations from 2010 to 2016 were included in the analysis. Personal identifiers were removed when data were released for the research. Informed consent documents were provided for participants to sign and let them make the decision to volunteer for the study. Ethics approval was approved by the Institutional Review Board (IRB) of the Tri-Service General Hospital and the MJ Health Management Institution.

### 2.2. Sperm Quality Analysis

Participants contributed their semen samples by masturbation after they had abstained for at least 3 days. Samples were stored in sterile containers that liquefied at 37 °C for 20 min and were sent to the laboratory for analysis [15]. Four parameters of sperm quality, including sperm concentration, total motility, progressive motility, and normal morphology, were recorded. A microcell counting chamber and a phase contrast microscope were used to assess sperm concentration. Sperm motility was classified as total motility and progressive motility based on the WHO 2010 classification [16]. Two hundred sperm cells were categorized into four different grades: A, B, C, or D. Total motility was defined as A + B + C and progressive motility as A + B. According to the WHO criteria in 2010, a normal sample was defined as if 4% (or 5th centile) or more of the observed sperm has normal sperm morphology.

### 2.3. Serum Total Bilirubin Measurement

The levels of serum total bilirubin were measured by a timed end-point Diazo method, using automatic biochemistry profiling (Beckman Synchron LX20 Beckman Coulter Inc., Fullerton, CA, USA). The analytical range and the reference range are 0.1~30 mg/dL and 0.2 to 1.3 mg/dL, respectively.

### 2.4. Covariates

A history of hypertension (HTN) and type II diabetes mellitus (DM) was obtained from a self-reported questionnaire. A question “How many packs do you smoke per day?” was used to determine the cigarette smoking status of participants. Systolic blood pressure (SBP) was measured by a standard sphygmomanometer when the subjects sat down. To collect the blood samples for analyzing laboratory data, including fasting plasma glucose (FPG), aspartate aminotransferase (AST), total cholesterol (CHO), and C-reactive protein (CRP), participants had to fast for at least 8 h, and these samples were measured by standard procedures.

### 2.5. Statistical Analyses

Associations between serum total bilirubin and sperm quality were performed using multivariable linear regression. The associations between various quartiles and the presence of sperm quality were analyzed by logistic regression. These regressions were adjusted by multivariable models, as follows. Model 1 was unadjusted. Model 2 included Model 1 and age. Model 3 included Model 2, ALT, Cr, and CRP. Model 4 included Model 3, a history of HTN, DM, and cigarette smoking. A significant difference was defined as a *p*-value of ≤0.05. Analyses in the current study were conducted using the Statistical Package for the Social Sciences, version 18.0 (SPSS Inc., Chicago, IL, USA) for Windows.

## 3. Results

### 3.1. Characteristics of Participants in Serum Total Bilirubin Quartiles

The general characteristic information of the 9057 subjects is listed in Table 1. The mean age of these quartiles was Q1: 32.33 ± 4.67, Q2: 32.35 ± 4.89, Q3: 32.01 ± 4.55, and Q4: 32.01 ± 4.79 years. The concentrations of serum total bilirubin of these quartile groups were 0.56 ± 0.10, 0.79 ± 0.06, 1.01 ± 0.07, and 1.50 ± 0.38, respectively. Participants in the highest quartile had significantly lower levels of FPG, CHO, and CRP and higher prevalence of cigarette smoking (*p* < 0.05). Sperm parameters, such as sperm concentration, motility, progressive motility, and normal morphology, showed significant differences across these quartiles (*p* < 0.05).

### 3.2. Associations between Serum Total Bilirubin and Sperm Quality

Associations between serum total bilirubin and sperm quality are shown in Table 2. Serum total bilirubin was significantly associated with decreased sperm motility and sperm normal morphology with β of −13.82 (95%CI: −26.99, −0.64) and −18.38 (95%CI: −30.46, −6.29). However, no significant difference was noted in the other sperm quality parameters.

### 3.3. Associations between Serum Total Bilirubin Quartiles and Sperm Quality

In Table 3, we divided serum total bilirubin into quartiles and analyzed the associations between these quartiles and sperm quality. The highest quartile of serum total bilirubin was inversely associated with sperm motility and sperm normal morphology with a β of −14.15 (95%CI: −28.36, 0.06) and −21.15 (95%CI: −33.99, −8.30).

## 4. Discussion

In this study, we highlighted the relationship between serum total bilirubin and sperm quality in a cross-sectional study. Serum total bilirubin had an inverse association with sperm motility and normal morphology. The highest quartile of serum total bilirubin was associated with decreased sperm motility and normal morphology. To the best of our knowledge, our study is the first to examine the association between serum total bilirubin and sperm quality in a reproductive-age male adult population.

The impacts of unconjugated bilirubin (UCB) on neurotoxicity have been broadly studied, and impairment of the cell membrane function, structure, and property are involved [17,18]. Generally, bilirubin is amphipathic but is lipophilic to cell membranes [19,20]. Apparently, it is possible that bilirubin can cross the blood–brain barrier (BBB) and enter the brain [21]. Bilirubin toxicity to brain cells might involve several plausible features, including induction of cell death, production of oxidative stress, and pro-inflammatory cytokines [22,23,24]. In a review study, Brito et al. reported that UCB affected brain microvascular endothelial cells, which play important roles in the maintenance of a functional BBB, and then caused hyperbilirubinemia-induced brain damage [25]. The interaction between bilirubin and spermatogenesis has not yet been reported in previous studies. The blood-testis barrier (BTB) might be the plausible target that explains how serum total bilirubin impacts sperm quality. Germ cell development is related to BTB because Sertoli cells can influence the chemical composition of the luminal fluid by controlling the adluminal compartment [26]. BTB also prevents germ cells from blood-borne noxious agents and inhibits cytotoxic agents into the seminiferous tubules [27]. Collectively, this barrier is the first line to protect the reproductive circulation system from toxic molecules. It is possible that bilirubin might lead to altered sperm quality by affecting BTB through several pathways.

The protective function of efflux transporters of Sertoli cells has been reported to prevent germ cells from toxic exposure [28,29]. Multidrug resistance-related protein (MRP), an efflux transporter that is detected mostly in Sertoli cells and Leydig cells of humans and mice [30,31], is known to transport a wide range of hydrophilic anion conjugates, hydrophobic xenobiotics, and natural compounds [32]. In a systemic review, MRP is suggested to be an important transporter of bilirubin that mediates ATP-dependent cellular export of bilirubin and has a protective effect against bilirubin-induced cytotoxicity [33]. The impact of bilirubin on spermatogenesis might be through this pathway.

Breast cancer resistance protein (BCRP) is an important efflux transporter that limits substances in the brain [34,35]. Xu et al. demonstrated that UCB elevation impaired the expression of BCRP at the BBB, which led to hepatic encephalopathy in vivo [36]. BCRP was observed in Sertoli cells, which was consistent with its localization at the BTB [37]. Organic anion-transporting polypeptides (OATPs) are influx pumps for a wild range of endogenous xenobiotics and compounds [38]. Steeg et al. indicated that OATP transporters played a crucial role in the uptake of unconjugated bile acids and drugs and hepatic reuptake of conjugated bilirubin [39]. Previous studies have reported that some of these uptake transporters are expressed only by spermatogonia, Sertoli cells, and BTB [40,41]. These membrane transporters might explain the impact of bilirubin on impaired sperm quality.

There are still several limitations to the present study. First, the cross-sectional design does not permit causal inference of the relationship between serum total bilirubin and sperm quality. A cohort analysis is needed in further studies. Second, we excluded individuals who might lead to selection bias and unsatisfactory generalization to avoid the confounding effects of hepatobiliary disease and to minimize age misclassification. Third, only one semen measurement was conducted for most participants, which limited us from having a repeated measurement, which might lead to within-person variations over time. Next, information on the participants’ medical history associated with bilirubin metabolism, such as hepatitis and bile duct obstruction, was lacking from the database. Last, laboratory data about potential confounding biomarkers, such as direct and indirect bilirubin, were not available from the MJ dataset. These biomarkers may be involved in the relationship between serum bilirubin and sperm quality by different mechanisms.

## 5. Conclusions

In this cross-sectional study, we highlighted that serum total bilirubin was negatively associated with sperm motility and normal morphology in an adult population. Although the causality of serum bilirubin to sperm quality must be established in a further prospective study, our finding provides epidemiological evidence for plausible interventional strategies for developing public health messages for men considering fatherhood.

## Figures and Tables

**Table 1 toxics-10-00295-t001:** Characteristics of participants in serum total bilirubin quartiles.

Variables	Q1 (*n* = 2298)	Q2 (*n* = 2230)	Q3 (*n* = 2277)	Q4 (*n* = 2252)	*p*-Value
Continuous variables, mean (SD)					
Age (years)	32.33 (4.67)	32.35 (4.89)	32.01 (4.55)	32.01 (4.79)	<0.05
Total bilirubin	0.56 (0.10)	0.79 (0.06)	1.01 (0.07)	1.50 (0.38)	<0.05
SBP	119.35 (14.51)	120.76 (13.04)	120.72 (11.65)	118.81 (13.67)	0.74
FPG	98.52 (12.11)	98.29 (15.34)	97.59 (14.30)	96.60 (11.82)	<0.05
AST	24.35 (11.53)	24.89 (11.51)	25.29 (17.24)	24.94 (18.46)	0.21
CHO	190.39 (34.52)	192.53 (35.37)	190.97 (38.00)	187.84 (32.82)	<0.05
CRP (mg/dL)	0.28 (0.51)	0.20 (0.30)	0.20 (0.62)	0.18 (0.37)	<0.05
Continuous variables, median (IQR)					
Sperm concentration	49.78 (36.33)	53.48 (42.29)	55.46 (44.13)	53.94 (41.20)	<0.05
Sperm total motility (%)	64.07 (16.16)	64.71 (16.65)	64.97 (15.67)	63.66 (16.71)	<0.05
Sperm progressive motility (%)	45.49 (16.56)	46.42 (17.27)	46.53 (16.60)	45.29 (16.92)	<0.05
Sperm normal morphology (%)	66.16 (16.69)	66.91 (16.44)	67.55 (15.83)	67.50 (16.86)	<0.05
Category variables, (%)					
HTN (%)	54 (2.4)	53 (2.4)	47 (2.1)	58 (2.8)	0.80
DM (%)	13 (0.6)	12 (0.5)	13 (0.6)	6 (0.3)	0.19
Cigarette smoking (%)	239 (40.9)	266 (50.2)	304 (52.6)	364 (58.4)	<0.05

SD, standard deviation; SBP, systolic blood pressure; FPG, fasting plasma glucose; AST, aspartate aminotransferase; CHO, cholesterol; CRP, C-reactive protein; HTN, hypertension; DM, type II diabetes mellitus.

**Table 2 toxics-10-00295-t002:** Association between serum total bilirubin and sperm quality.

Variables	Model 1 ^a^ β (95% CI)	*p* Value	Model 2 ^a^ β (95% CI)	*p* Value	Model 3 ^a^ β (95% CI)	*p* Value	Model 4 ^a^ β (95% CI)	*p* Value
	Sperm Concentration							
Total bilirubin	−6.40 (−35.10, 22.31)	0.66	1.44 (−26.60, 29.47)	0.92	3.50 (−25.58, 32.57)	0.81	3.02 (−27.80, 33.85)	0.84
	Sperm Motility							
Total bilirubin	−7.41 (−20.68, 5.86)	0.27	−12.11 (−24.45, 0.23)	<0.05	−12.13 (−24.74, 0.47)	<0.05	−13.82 (−26.99, −0.64)	<0.05
	Sperm Progressive Motility							
Total bilirubin	−8.28 (−22.25, 5.69)	0.24	−12.06 (−25.71, 1.60)	0.08	−10.91 (−25.21, 3.38)	0.13	−12.73 (−27.70, 2.24)	0.09
	Sperm Normal Morphology							
Total bilirubin	−16.75 (−27.53, −5.96)	<0.05	−17.89 (−29.01, −6.76)	<0.05	−17.46 (−28.95, −5.97)	<0.05	−18.38 (−30.46, −6.29)	<0.05

^a^ Adjusted covariates: Model 1: unadjusted. Model 2: Model 1 + age. Model 3: Model 2 + AST, CHO, CRP, FPG, and SSL. Model 4: Model 3 + HTN, cigarette smoking.

**Table 3 toxics-10-00295-t003:** Association between quartiles of serum total bilirubin and sperm quality.

Variables	Model 1 ^a^ OR ^b^ (95% CI)	*p* Value	Model 2 ^a^ OR ^b^ (95% CI)	*p* Value	Model 3 ^a^ OR ^b^ (95% CI)	*p* Value	Model 4 ^a^ OR ^b^ (95% CI)	*p* Value
	Sperm Concentration							
Q2 vs. Q1	7.13 (−25.46, 39.72)	0.66	7.78 (−23.25, 38.81)	0.62	11.55 (−20.55, 43.65)	0.47	11.98 (−20.98, 44.94)	0.47
Q3 vs. Q1	9.06 (−20.88, 39.00)	0.55	12.57 (−16.09, 41.24)	0.38	23.91 (−7.61, 55.43)	0.13	25.76 (−7.26, 58.77)	0.12
Q4 vs. Q1	−9.80 (−42.39, 22.79)	0.55	−2.14 (−33.87, 29.59)	0.89	−2.81 (−34.93, 29.30)	0.86	−4.18 (−37.26, 28.91)	0.80
	Sperm Motility							
Q2 vs. Q1	2.67 (−12.49, 17.83)	0.72	2.26 (−11.39, 15.91)	0.74	5.14 (−8.77, 19.05)	0.46	5.25 (−8.90, 19.40)	0.46
Q3 vs. Q1	0.74 (−13.19, 14.67)	0.92	−1.45 (−14.06, 11.16)	0.82	2.23 (−11.43, 15.89)	0.74	3.67 (−10.51, 17.84)	0.60
Q4 vs. Q1	−7.93 (−23.09, 7.23)	0.30	−12.72 (−26.68, 1.24)	0.07	−13.31 (−27.22, 0.61)	0.06	−14.15 (−28.36, 0.06)	<0.05
	Sperm Progressive Motility							
Q2 vs. Q1	−5.53 (−21.64, 10.57)	0.49	−5.85 (−21.20, 9.50)	0.45	−3.67 (−20.01, 12.67)	0.65	−3.72 (−20.42, 12.98)	0.65
Q3 vs. Q1	−4.01 (−18.81, 10.78)	0.59	−5.73 (−19.91, 8.45)	0.42	−2.56 (−18.61, 13.49)	0.75	−0.99 (−17.72, 15.74)	0.91
Q4 vs. Q1	−9.83 (−25.94, 6.27)	0.23	−13.58 (−29.28, 2.11)	0.09	−13.22 (−29.57, 3.12)	0.11	−13.99 (−30.76, 2.77)	0.10
	Sperm Normal Morphology							
Q2 vs. Q1	−7.32 (−19.68, 5.05)	0.24	−7.40 (−19.84, 5.03)	0.24	−4.78 (−17.30, 7.75)	0.44	−5.04 (−17.84, 7.75)	0.43
Q3 vs. Q1	−3.81 (−15.17, 7.55)	0.50	−4.28 (−15.76, 7.21)	0.46	0.46 (−11.83, 12.76)	0.94	1.65 (−11.17, 14.46)	0.80
Q4 vs. Q1	−18.62 (−30.98, −6.25)	<0.05	−19.64 (−32.35, −6.92)	<0.05	−20.79 (−33.32, −8.26)	<0.05	−21.15 (−33.99, −8.30)	<0.05

^a^ Adjusted covariates: Model 1: unadjusted, Model 2: Model 1 + age, Model 3: Model 2 + AST, CHO, CRP, FPG, SSL, Model 4: Model 3 + HTN, cigarette smoking. β ^b^ was interpreted as change of sperm quality for each increase in total bilirubin.

## Data Availability

The datasets analyzed during the current study are not publicly available due to but are available from the corresponding author on reasonable request.

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
