# Peer review of "Serum Bilirubin and Sperm Quality in Adult Population"

_toxics, 2022, doi:10.3390/toxics10060295_

Round 1

Reviewer 1 Report

This paper focuses on the relationship between sperm quality and bilirubin. Research on sperm is limited and the findings need from this perspective. However, the following points need to be improved.

METHOD.
I did not know why the authors identified DM and history of hypertension as covariates. When choosing a covariate, one should choose a variable that affects both the explanatory and the objective variables. It is unclear what DAG the authors considered and how they constructed their statistical model. It should also be stated whether any history of hepatitis, bile duct obstruction, hemolytic anemia, etc. that could affect the bilirubin level was checked.

Temperature control of sperm after collection is important, which is also mentioned in the citation, but it should be clearly stated how this was done in this study. Furthermore, if it is based on citation 15, wouldn't the most recent version be more appropriate
https://apps.who.int/iris/bitstream/handle/10665/343208/9789240030787-eng.pdf?sequence=1&isAllowed=y

RESULT.
Regarding serum total bilirubin, it is not clear whether the target population of this paper is in the normal range or whether values including abnormal values were included. Because this is an important point, descriptive statistics such as the median (percentile) or mean (SD) of serum total bilirubin for this population should be added to the text.

We would like to confirm the following statement and the statement in the footnote of Table 2.
The reference value for bilirubin is in the range of approximately 0.2 to 1.2 mg/dL. It is unclear in which range this population was in but is it correct to interpret that when the bilirubin level increases from 1 mg/dL to 2 mg/dL, the sperm motility decreases by 13.8%, and the sperm normal morphology decreases by 18.38%. As far as we can see from Table 1, the difference in Median values of sperm motility and sperm normal morphology for each bilirubin quartile is only a few percent. This should be improved to make it easier for readers to understand.

The levels of serum total bilirubin were significantly associated with decreased sperm motility and percentage of sperm normal morphology with β of -13.82 (95%CI: -26.99, 0.64) and -18.38 (95%CI: -30.46, -6.29). 

Beta was interpreted as a change in sperm quality for each increase in total bilirubin

DISCUSSION.
The study limitations are described, but the levels of direct and indirect bilirubin are not known, so it is difficult to discuss the relationship between sperm quality and bilirubin in detail, but the more background and discussion of why bilirubin was thought to affect sperm quality is needed.

Author Response

Reviewer 1

This paper focuses on the relationship between sperm quality and bilirubin. Research on sperm is limited and the findings need from this perspective. However, the following points need to be improved.

METHOD.
1. I did not know why the authors identified DM and history of hypertension as covariates. When choosing a covariate, one should choose a variable that affects both the explanatory and the objective variables. It is unclear what DAG the authors considered and how they constructed their statistical model. It should also be stated whether any history of hepatitis, bile duct obstruction, hemolytic anemia, etc. that could affect the bilirubin level was checked.

Response: Thank you for your thorough review and salient observations. Numerous evidence suggests that serum total bilirubin is associated with both diabetes and hypertension. Serum bilirubin concentrations were positively associated with the risk of incident diabetes. Total bilirubin was reported to have protective effect on the risk of hypertension2. Due to this condition, we included history of diabetes and hypertension in adjustment.

Reference:

  1. Wang, J., Li, Y., Han, X. et al.Serum bilirubin levels and risk of type 2 diabetes: results from two independent cohorts in middle-aged and elderly Chinese. Sci Rep7, 41338 (2017).
  2. Stec DE, Hosick PA, Granger JP. Bilirubin, renal hemodynamics, and blood pressure. Front Pharmacol. 2012;3:18. Published 2012 Feb 14. doi:10.3389/fphar.2012.00018

In addition, the information of hepatitis, bile duct obstruction, etc. is lacking from the database. We have added this issue into limitation section.

Next, the information of participants’ medical history associated with bilirubin metabolism, such as hepatitis and bile duct obstruction, was lacking from the database.

2. Temperature control of sperm after collection is important, which is also mentioned in the citation, but it should be clearly stated how this was done in this study. Furthermore, if it is based on citation 15, wouldn't the most recent version be more appropriate
https://apps.who.int/iris/bitstream/handle/10665/343208/9789240030787-eng.pdf?sequence=1&isAllowed=y

Response: Thank you for your thorough review and salient observations. We have revised the sentence and changed the citation in Method section based on your recommendation.

Samples were stored in sterile containers which liquefied 37°C for 20 minutes and sent to the laboratory for analysis1.

Reference:

  1. Organization WH. Manual for the Examination of Human Semen and Semencervical Mucus Interaction. 5th edition. New York: NY, USA: Cambridge University Press. 2010.

RESULT.
3. Regarding serum total bilirubin, it is not clear whether the target population of this paper is in the normal range or whether values including abnormal values were included. Because this is an important point, descriptive statistics such as the median (percentile) or mean (SD) of serum total bilirubin for this population should be added to the text.

Response: Thank you for your thorough review and salient observations. We have added the levels of serum total bilirubin into Table 1 and revised the Result section based on your recommendation.

The concentration of serum total bilirubin of these quartile groups was 0.56±0.10, 0.79±0.06, 1.01±0.07, and 1.50±0.38, respectively.

4. We would like to confirm the following statement and the statement in the footnote of Table 2. The reference value for bilirubin is in the range of approximately 0.2 to 1.2 mg/dL. It is unclear in which range this population was in but is it correct to interpret that when the bilirubin level increases from 1 mg/dL to 2 mg/dL, the sperm motility decreases by 13.8%, and the sperm normal morphology decreases by 18.38%. As far as we can see from Table 1, the difference in Median values of sperm motility and sperm normal morphology for each bilirubin quartile is only a few percent. This should be improved to make it easier for readers to understand.

The levels of serum total bilirubin were significantly associated with decreased sperm motility and percentage of sperm normal morphology with β of -13.82 (95%CI: -26.99, 0.64) and -18.38 (95%CI: -30.46, -6.29). 

Beta was interpreted as a change in sperm quality for each increase in total bilirubin

Response: Thank you for your thorough review and salient observations. We have removed the footnote and revised the manuscript based on your recommendation.

Associations between serum total bilirubin and sperm quality were shown in Table 2. Serum total bilirubin were significantly associated with decreased sperm motility and sperm normal morphology with β of -13.82 (95%CI: -26.99, -0.64) and -18.38 (95%CI: -30.46, -6.29). However, no significant difference was noted in other sperm quality parameters.

DISCUSSION.
5. The study limitations are described, but the levels of direct and indirect bilirubin are not known, so it is difficult to discuss the relationship between sperm quality and bilirubin in detail, but the more background and discussion of why bilirubin was thought to affect sperm quality is needed.

Response: Thank you for your thorough review and salient observations. We have added the levels of total bilirubin into Result section and revised the Discussion section based on your recommendation.

The concentration of serum total bilirubin of these quartile groups was 0.56±0.10, 0.79±0.06, 1.01±0.07, and 1.50±0.38, respectively.

The impacts of unconjugated bilirubin (UCB) on neurotoxicity have been broadly studied that impairment of the cell membrane function, structure, and property are involved. Generally, bilirubin is amphipathic but is lipophilic to cell membranes. Apparently, it is possible that bilirubin can cross blood-brain barrier (BBB) and enter the brain. Bilirubin toxicity to brain cells might involve several plausible features including induction of cell death, production of oxidative stress, and pro-inflammatory cytokines. In a review study, Brito et al. reported that UCB affected brain microvascular endothelial cells, which are important roles in the maintenance of a functional BBB, then caused hyperbilirubinemia-induced brain damage. The interaction between bilirubin and spermatogenesis has not been reported in previous studies yet. The blood-testis barrier (BTB) might be the plausible target that explains how serum total bilirubin impacts on sperm quality. Germ cell development is related to BTB because Sertoli cells can influence the chemical composition of the luminal fluid by controlling the adluminal compartment. BTB also prevents the germ cells from blood-borne noxious agents and inhibits cytotoxic agents into the seminiferous tubules. Collectively, this barrier is the first line to protect reproductive circulation system from toxic molecules. It is possible that bilirubin might lead to altered sperm quality by affecting BTB with several pathways.

The protective function of efflux transporters of Sertoli cells has been reported that might prevent the germ cells from toxic exposure. Multidrug resistance- related protein (MRP), an efflux transporter which is detected mostly in Sertoli cells and Leydig cells of humans and mice, is known to transport a wide range of hydrophilic anion conjugates, hydrophobic xenobiotics, and natural compounds. In a systemic review, MRP is suggested to be an important transporter of bilirubin that mediates ATP-dependent cellular export of bilirubin and has protective effect against bilirubin-induced cytotoxicity. The impact of bilirubin on spermatogenesis might be through this pathway.

Breast cancer resistance protein (BCRP) is an important efflux transporter that limits substances into the brain. Xu et al. demonstrated that UCB elevation impaired the expression of BCRP at the BBB that led to hepatic encephalopathy in vivo. BCRP was observed in Sertoli cells that was consistent with its localization at the BTB. Organic anion transporting polypeptides (OATPs) are influx pumps for a wild range of endogenous xenobiotics and compounds. Steeg et al. indicated that OATP transporters played a crucial role in uptake of unconjugated bile acids and drugs and hepatic reuptake of conjugated bilirubin. Previous studied have reported that some of these uptake transporters expressed only by spermatogonia, Sertoli cell, and BTB. These membrane transporters might explain the impact of bilirubin on impaired sperm quality.

Reviewer 2 Report

Dear Authors,

In the available resources of online journal databases, one can find studies that analysed the relationship between semen quality and the presence of lifestyle stressors such as occupational stress, life events or infertility of the couple. They provide specific evidence of deterioration of semen quality under the influence of psychological stress. In the papers reviewed, the authors examined the relationship between serum total bilirubin levels and semen quality in an adult population. The topic is very interesting, nevertheless its presentation has given me some suggestions/observations which I include below:
Line 3: shouldn't "and Ph.D." be removed from the list of authors of the manuscript?
Line 13: there seems to be too much data given in the author information for correspondence, especially as it is duplicated (affiliation is given above).
Line 17, 20, 29 - please remove words, i.e. background, method, conclusions.
Line 36, 37, 38 etc. - no spaces before the citation record: _[ ] in multiple lines. Please correct.
Line 34: Too laconic introduction to the research area covered.
Line 65: Too laconic description of the documentation for consent to conduct this type of research with human subjects.
Line 137: The discussion of results is also quite short, citing little information to understand why the research you conducted, learning about the mechanisms discussed in the adult population is so important.
Line 201: Please edit the literature items as required by the journal.

Author Response

Reviewer 2
In the available resources of online journal databases, one can find studies that analysed the relationship between semen quality and the presence of lifestyle stressors such as occupational stress, life events or infertility of the couple. They provide specific evidence of deterioration of semen quality under the influence of psychological stress. In the papers reviewed, the authors examined the relationship between serum total bilirubin levels and semen quality in an adult population. The topic is very interesting, nevertheless its presentation has given me some suggestions/observations which I include below:

1. Line 3: shouldn't "and Ph.D." be removed from the list of authors of the manuscript?
Response: Thank you for your thorough review and salient observations. We have revised the sentence based on your recommendation.

Yuan-Yuei Chen, M.D.1,2, Wei-Liang Chen, M.D. Ph.D. 3,4,5

2. Line 13: there seems to be too much data given in the author information for correspondence, especially as it is duplicated (affiliation is given above).
Response: Thank you for your thorough review and salient observations. We have revised the correspondence based on your recommendation.

3. Line 17, 20, 29 - please remove words, i.e. background, method, conclusions.
Response: Thank you for your thorough review and salient observations.The neurotoxicity of bilirubin has been extensively reported in numerous studies. However, the association between bilirubin and male fertility has not been studied yet. The main goal of the study was to investigate the association between serum total bilirubin and sperm quality in an adult population. 9057 participants who attended the MJ health examination (2010-2016) were enrolled in this cross-sectional study. Sperm specimen was collected by masturbation and sperm quality was analyzed in accordance to the WHO criteria. Levels of serum total bilirubin were measured by an automatic biochemical profile analyzer. Thereafter, the associations between serum total bilirubin and sperm quality were performed by a multivariable linear regression. Serum total bilirubin was inversely associated with sperm concentration and normal morphology with β values of -13.82 (95% CI: -26.99, -0.64) and -18.38 (95% CI: -30.46, -6.29) after adjusting for covariables. The highest levels of serum total bilirubin were significantly associated with sperm concentration and normal morphology with β values of -14.15 (95% CI: -28.36, 0.06) and -21.15 (95% CI: -33.99, -8.30). Our study highlighted the potential impact of serum bilirubin on sperm quality in a male population. Additional longitudinal researches are necessary to explore these findings and underlying mechanisms.

4. Line 36, 37, 38 etc. - no spaces before the citation record: _[ ] in multiple lines. Please correct.

Response: Thank you for your thorough review and salient observations. We have revised these problems in the whole manuscript based on your recommendation.

5. Line 34: Too laconic introduction to the research area covered.
Response: Thank you for your thorough review and salient observations. We have revised the Introduction section based on your recommendation.

6. Line 65: Too laconic description of the documentation for consent to conduct this type of research with human subjects.
Response: Thank you for your thorough review and salient observations. We have revised the sentence based on your recommendation.

Personal identifiers were removed when data release for the research. Informed consent documents were provided for participants to sign and let them make decision to volunteer for study. Ethics approval was approved by the Institutional Review Board (IRB) of the Tri-Service General Hospital and the MJ Health Management Institution.

7. Line 137: The discussion of results is also quite short, citing little information to understand why the research you conducted, learning about the mechanisms discussed in the adult population is so important.
Response: Thank you for your thorough review and salient observations. We have revised the discussion based on your recommendation.

8. Line 201: Please edit the literature items as required by the journal.

Response: Thank you for your thorough review and salient observations. We have revised the manuscript based on your recommendation.
